# Gene Targeting to the Cerebral Cortex Following Intranasal Administration of Polyplexes

**DOI:** 10.3390/pharmaceutics14061136

**Published:** 2022-05-26

**Authors:** Asya I. Petkova, Ilona Kubajewska, Alexandra Vaideanu, Andreas G. Schätzlein, Ijeoma F. Uchegbu

**Affiliations:** 1UCL School of Pharmacy, 29–39 Brunswick Square, London WC1N 1AX, UK; asya.petkova.15@ucl.ac.uk (A.I.P.); i.kubajewska@ucl.ac.uk (I.K.); a.vaideanu@ucl.ac.uk (A.V.); a.schatzlein@ucl.ac.uk (A.G.S.); 2Nanomerics Ltd., Northwick Park and St. Mark’s Hospital, Y Block, Watford Road, London HA1 3UJ, UK

**Keywords:** nose to brain delivery, gene therapy, polyplexes, hyaluronidase, glycol chitosan

## Abstract

Gene delivery to the cerebral cortex is challenging due to the blood brain barrier and the labile and macromolecular nature of DNA. Here we report gene delivery to the cortex using a glycol chitosan—DNA polyplex (GCP). In vitro, GCPs carrying a reporter plasmid DNA showed approximately 60% of the transfection efficiency shown by Lipofectamine lipoplexes (LX) in the U87 glioma cell line. Aiming to maximise penetration through the brain extracellular space, GCPs were coated with hyaluronidase (HYD) to form hyaluronidase-coated polyplexes (GCPH). The GCPH formulation retained approximately 50% of the in vitro hyaluronic acid (HA) digestion potential but lost its transfection potential in two-dimensional U87 cell lines. However, intranasally administered GCPH (0.067 mg kg^−1^ DNA) showed high levels of gene expression (IVIS imaging of protein expression) in the brain regions. In a separate experiment, involving GCP, LX and naked DNA, the intranasal administration of the GCP formulation (0.2 mg kg^−1^ DNA) resulted in protein expression predominantly in the cerebral cortex, while a similar dose of intranasal naked DNA led to protein expression in the cerebellum. Intranasal LX formulations did not show any evidence of protein expression. GCPs may provide a means to target protein expression to the cerebral cortex via the intranasal route.

## 1. Introduction

Delivering intravenously injected macromolecules, such as genes, to the brain is severely hampered by the blood brain barrier (BBB) [1] While the BBB is a major obstacle when delivering macromolecules to the brain, the delivery of genes to the brain is further hampered by the fact that gene delivery nanoparticles (NPs) are rapidly cleared from the circulation due to the interaction between the positively charged NPs and the negatively charged plasma proteins and erythrocytes. These electrostatic interactions result in aggregation, opsonization and subsequent clearance of the particles from the body [2].

Intranasal administration is used to treat nasal epithelium infections or conditions such as nasal congestion, rhinorrhoea and rhinosinusitis [3]. More recently, the potential of intranasal delivery to ensure transport through the olfactory bulb to the brain, bypassing the BBB, has been explored for a variety of molecules [4,5,6,7]. Additionally, the intranasal delivery route has other advantages such as ease of administration, non-invasiveness, rapid onset of action, a relatively large and permeable surface area and avoidance of the first-pass hepatic metabolism. Intranasally administered insulin has shown promise in alleviating the symptoms in both Alzheimer’s disease type (AD-type) dementia and non-AD-type dementia in clinical trials, resulting in improvement of memory, cognitive functions and attention [8,9,10]. In parallel with these exciting clinical results, there are also a number of studies in animal models describing the promise of intranasally administered lectin, for the treatment of obesity, and oxytocin, as a treatment option for depression [11,12,13,14] In terms of gene delivery, the intranasal route was first used for the delivery of a reporter plasmid to the brain. A 30mer peptide with PEGylated lysine residues was shown to deliver a green fluorescent protein (GFP) DNA, resulting in GFP expression in multiple brain regions, predominantly in the frontal cortex [15]. Similarly, intranasally administered PEI and chitosan-coated superparamagnetic iron oxide nanoparticles (SPIONs) were shown to deliver a reporter plasmid coding for a red fluorescent protein (RFP) to the cortex and hippocampus of rats. Intranasally delivered NPs formed by DNA coding for glial cell line-derived neurotrophic factor (GDNF) complexed with polyethylene glycol (substituted with lysine residues) showed peak expression in the rat striatum a week after administration and neuroprotective action in a rat model of Parkinson’s disease (PD) [16]. Similarly, a self-assembling electrostatic complex between an antisense RNA against a microRNA (miR-21) associated with inhibition of pro-apoptotic genes and a peptide delivered intranasally to mice bearing intracranial tumours showed a significant decrease in the tumour volume a week post administration of the NPs [17]. Furthermore, miR-21 levels were reduced, while the expression of pro-apoptotic genes such as PTEN and PDCD4 was induced by the NPs.

Nose to brain delivery is well-documented in humans [18] and has been reviewed recently [19]. Molecules transported via the intranasal route of administration after mucociliary clearance reach the interior of the nasal cavity where the respiratory, olfactory neuronal networks and blood vessels are accessible [20,21]. The neuronal transport pathway involves the transport of molecules from the nasal cavity to the brain parenchyma and the pons along the olfactory and trigeminal nerves, respectively. The movement of molecules from the nasal cavity to the olfactory bulb is facilitated through intracellular or extracellular pathways. A very slow axonal transport from the olfactory bulb and other brain regions is performed by passive diffusion or receptor-mediated endocytosis, both facilitating intracellular trafficking in olfactory neurons [22]. By contrast, the extracellular transport of molecules from the nasal cavity to the brain is a much more rapid process. Rapid delivery, almost immediately or up to one hour after intranasal administration [23,24,25,26], has been reported for many molecules, hence an extracellular pathway is the most likely mode of transport in those instances. Extracellular transport may play a role in the movement of peptide-loaded nanoparticles within the brain [27]. Lochhead et al. showed the presence of the tracer dextrans 20 min after nasal administration in the perivascular space of the nasal lamina propria and in different brain regions [28]. Similarly, fluorescently labelled insulin was found throughout the brain only 20 min after nasal administration [29]. These findings support the idea of rapid transport of substances from the olfactory bulb throughout the brain by the perivascular pathway. The perivascular transport hypothesis is also supported for the delivery of NPs loaded with a plasmid encoding for hGDNF, where the resulting transfected cells appear to be situated in the perivascular spaces surrounding capillary endothelial cells and are most likely pericytes [30].

While others have reported delivery to multiple brain regions [15,31] upon nasal administration, we aimed to achieve targeted delivery to the cortex in order to exploit the clinical potential for developing effective gene therapies against Alzheimer’s disease, posterior cortical atrophy and frontal lobe glioblastomas. Additionally, assuming an extracellular pathway, we sought to examine the effect of the HYD coating of the polyplexes to facilitate brain delivery via the nasal route of administration. HYD has been used as a spreading agent for the subcutaneous route for over 50 years, enabling volumes in excess of the normal 2 mL [32] to be administered due to a loosening of the connective tissue by enzymatic cleavage of ECM biopolymer components. Specifically, hyaluronidase—whether of bacterial or vertebral origin—catalyses the hydrolysis of HA at the 1,4-glycosidic linkages [33]. Vertebrate HYD, as has been used in the current study (endo-β-acetyl-hexosaminidase), also catalyses the degradation of chondroitin and chondroitin-6-sulphate at the 1,4-glycosidic linkage, albeit at a slower rate than for hyaluronic acid. Furthermore, subcutaneous formulations containing HYD that allow rapid injection of increased dose volumes of 5 mL of trastuzumab have recently been approved for human use and more recently a 10 mL volume of subcutaneous rituximab has been administered using HYD [34]. We have previously shown that HYD coating on nanoparticles has a beneficial effect on drug bioavailability when HYD nanoparticles are administered via the subcutaneous route; increasing plasma exposure by two-fold and increasing tumoricidal activity when compared to formulations devoid of HYD [35]. To the best of our knowledge, we have not seen reports of HYD coated nanoparticles being used for nose to brain delivery.

## 2. Materials and Methods

CellTiter 90^®^ AQ one solution cell proliferation assay, pSV-40 β-Galactosidase control vector, Beta-Glo^®^ and pGL4.13[luc2/SV40] vector, were supplied by Promega (Southampton, UK). Ninety-six well Corning cell culture plates, hydrochloric acid (purity ≥ 98%) were supplied by VWR (Fontenay-sous-Bois, France) and Luciferin was supplied by Perkin Elmer (Waltham, MA, USA). T-PER™ tissue protein extraction reagent, EDTA free halt protease inhibitor cocktail, Lipofectamine 2000 and acetonitrile HPLC grade (purity ≥ 99.5%) were supplied by Thermo Scientific (Loughborough, UK). Visking dialysis tubing was supplied by Medicell Membranes Ltd. (London, UK). Glycol chitosan Mw = 113 kDa, Mn = 98 kDa was supplied by Wako (Osaka, Japan). Deuterium oxide, sodium acetate anhydrous (purity ≥ 99%), glacial acetic acid, hyaluronidase Type I S-400–1000 units mg^−1^, trifluoracetic acid (purity ≥ 99%), hyaluronic acid (Mw 70–90 kDa), bovine serum albumin, Triton™ X-100, magnesium chloride hexahydrate (purity ≥ 96%), trypsin (0.02% EDTA, Gibco™ GlutaMAX, penicillin/streptomycin, minimal essential medium eagle (MEME), OptiMEM and foetal bovine serum were all supplied by Sigma Aldrich (Gillingham, UK).

### 2.1. Acid Degradation of Glycol Chitosan (GC)

Glycol chitosan (10 g) was dissolved in HCl (4 M, 375 mL) as described previously [36]. The flask was incubated for 2 h in a preheated water bath at 50 °C. Dialysis against water was performed over 24 h with 5–6 water changes (3.5 kDa MWCO). The dialyzed solution was freeze-dried on a Christ 1–4 LD plus freeze dryer (Martin Christ, Osterode am Harz, Germany).

### 2.2. Characterization of Glycol Chitosan (GC)

The samples for NMR analysis were prepared at 20 mg mL ^−1^ in deuterium oxide and analysed on either an AMX 400 MHz or an AMX 500 MHz spectrometer (Bruker, Rheinstetten, Germany). Molecular weight (Mw) measurements were performed on a GPC-MALLS dRI with a MALLS 120 mW solid-state laser (wavelength, λ658 nm) DAWN^®^ HELEOSTM and Optilab rEX Interferometric Refractometer, respectively (Wyatt Technology Corporation, Santa Barbara, CA, USA). Size exclusion chromatography (SEC) was performed using PolySep™—GFC-P 4000 column (300 × 7.8 mm) protected by a PolySep™ (Phenomenex, Macclesfield, UK) with a GFG-P guard column (35 × 7.8 mm).

### 2.3. Preparation of Glycol Chitosan Polyplexes and Lipoplexes with Lipofectamine (GCP 1 and LX 1)

Nanocomplexes for in vitro studies were prepared at a 6 µg mL^−1^ concentration of β-Gal DNA in phosphate buffer (20 mM, pH = 6.8) or in OptiMEM for Lipofectamine in a total volume of 500 µL. Equal volumes of plasmid DNA (250 µL) and polymer or Lipofectamine in phosphate buffer (250 µL) at a polymer, β-Gal DNA mass ratio of 60:1 (GCP1) or a Lipofectamine, β-Gal DNA mass ratio of 2:1 (LX 1) were prepared. The β-Gal DNA solution was always added to the polymer dispersion followed by mixing with a pipette for 10 s. Lipofectamine formulations were prepared by following the manufacturer’s instructions. Complexation was performed over 24 h and at 4 °C for GCP 1 and 30 min at room temperature for LX 1.

The polyplexes and lipoplexes for the in vivo experiments were prepared as described above but at a β-Gal DNA concentration of 250 µg mL ^−1^ in a total volume of 16 µL (a dose of 0.2 mg kg ^−1^ for GCP 3 and LX 3) and at a β-Gal concentration of 84 µg mL^−1^ in a total volume of 16 µL (a dose of 0.067 mg kg ^−1^ for GCP 2 and LX 2).

### 2.4. Preparation of Hyaluronidase Coated Nanocomplexes (GCPH)

Polyplexes with a polymer to DNA weight ratio of 60:1 were prepared containing 6 μg mL^−1^ DNA in phosphate buffer (20 mM, pH = 6.8) in a total volume of 250 μL and stored for 24 h at 4 °C. Working solutions of hyaluronidase in NaCl (20 mM, pH adjusted to 12 with 0.1 M NaOH) were prepared. The hyaluronidase solutions (250 μL) were added to the polyplex dispersion described above (250 μL), dropwise under magnetic stirring for 30 min. Subsequently, the now formed ternary complexes (hyaluronidase, glycol chitosan and DNA) were stored for another 24 h at 4 °C. For the scaled-up formulations for in vivo experiments, the mass ratios of plasmid, polymer and enzyme were kept at 1:60:84.

GCPH 7 for in vivo administration was prepared at a luciferase DNA concentration of 84 µg mL^−1^ (16 µL). Equal volumes of polyplex (8 µL) and hyaluronidase (8 µL 14.2 mg mL^−1^) were mixed, as described above, followed by incubation at 4 °C for 24 h.

The name codes for the ternary complexes (GCPH) include numbers that represent the amount of hyaluronidase added in mg mL^−1^. As such, GCPH 7 indicates that the formulation contained hyaluronidase at a concentration of 7 mg mL^−1^. The ratio of GC to DNA was always at a weight ratio of 60:1.

### 2.5. Dynamic Light Scattering (DLS)

Size and zeta potential measurements were both performed in a reusable zeta cell on a Malvern Zetasizer Nano ZS machine (Malvern Panalytical, Malvern, UK). Size measurements were performed first, followed by a zeta potential measurement. Prior to the measurements, the instrument was checked with size and zeta potential standards.

### 2.6. Reversed-Phase High Performance Liquid Chromatography (RP-HPLC)

Hyaluronidase analysis was performed using a reverse-phase PRLP-S column (4.6 × 50 mm in length, pore size = 3 µm) and an RP-HPLC system (Agilent Technologies, Santa Clara, CA, USA). The system was fitted with a guard column and analysis was carried out at a flow rate of 0.7 mL min^−1^. The column temperature was set to 80 °C, the injection volume set at 10 μL and the wavelength at 280 nm. 0.1% Trifluoroacetic acid (TFA)/Acetonitrile (ACN) was used as the mobile phase and run at a gradient (0.50 min 10% ACN, 0.51 min 90% ACN, 4.00 min 90% ACN, 4.01 min 10% ACN). Samples were prepared in 0.1% TFA and analysed using a standard curve (y = 708.13x − 5.6753, r^2^ = 0.991).

### 2.7. Cell Culture

U-87 MG cells (ATCC^®^ HTB-14™) were maintained in 75 cm^2^ blue vent cap culture flasks in 10–12 mL of minimal essential medium eagle (MEME) supplemented with Sodium Pyruvate (1% *v*/*v*), GlutaMAX°TM (1% *v*/*v*), Penicillin/Streptomycin (1% *v*/*v*) and foetal bovine serum (10% *v*/*v*).

### 2.8. In Vitro Transfection Experiments

U-87 cells were seeded in lysine-coated 6 well plates at a density of 5 × 10^5^ cells per well in 1 mL of MEME. The cells were left for 72 h to reach the exponential growth phase and were then treated with the nanocomplexes or naked DNA. After 72 h the full MEME from each well of the plate was replaced by FBS free MEME (1.5 mL). An aliquot of the nanocomplexes (0.5 mL) was then added to each well containing FBS free medium MEME (1.5 mL) to make up the total volume to 2 mL per well. The plates were then left for 17 h in an incubator at 37 °C in the presence of 5% CO_2_. At the end of the incubation period, the FBS-free medium with the treatments was aspirated and replaced with 2 mL per well of full MEME containing FBS (10% *v*/*v*). The plates were then left in the incubator at 37 °C for another 24 h before performing the β-Galactosidase assay.

### 2.9. β-Galactosidase Assay

The β-Galactosidase enzyme assay was purchased from Promega (Southampton, UK) and was performed according to the manufacturer’s instructions.

### 2.10. MTS Assay

U-87 cells were seeded in 96 well plates at a density of 10^3^ cells per well in a total volume of 100 µL and left for 72 h to reach the exponential phase. Two-fold serial dilutions across the plate were performed for Lipofectamine at a starting concentration of 0.5 mg mL^−1^ Lipofectamine. Individual solutions were prepared for GC37 at the following concentrations (5 mg mL^−1^, 4.6 mg mL^−1^, 4.4 mg mL^−1^, 4.0 mg mL^−1^, 3.6 mg mL^−1^, 3.0 mg mL^−1^, 2.6 mg mL^−1^, 2.2 mg mL^−1^ and 1.8 mg mL^−1^). Cells were treated for 17 h in FBS free medium, then recovered for 24 h in complete medium. Five wells containing cells without any treatment were left as a negative control and 5 wells treated with Triton X 100 (0.05% *w*/*v*) were used as a positive control. The MTS reagent (20 µL) was added to all the wells and the absorbance at 490 nm was measured using a SpectraMax M series spectrophotometer (Molecular Devices, San Jose, CA, USA) after 2 h.

IC50 values were calculated from equations generated by fitting data plots to a linear regression model in GraphPad Prism.

### 2.11. Hyaluronic Acid Digestion Assay

Hyaluronic acid (70–90 kDa, 10 mg) was dissolved in phosphate buffer (0.3 M sodium phosphate, pH = 5.35). The solution was then heated to 90 °C with stirring until all the hyaluronic acid was dissolved, followed by cooling down to 37 °C in a water bath. The assay was performed according to the manufacturer’s instructions available at Enzymatic Assay of Hyaluronidase (3.2.1.35) (sigmaaldrich.com) [37].

### 2.12. Intranasal Dosing

All animal experiments were performed under a UK Home Office licence and in accordance with the UK Animal Scientific Procedures Act 1986 (ASPA). A local ethics committee approved the experimental procedures. The experiments were carried out in the biological safety unit at the UCL School of Pharmacy. Female, BALB/c mice weighing ~20 g (Charles River, Harlow, UK) were initially kept for a week to acclimatise prior to the start of the experimental work in the animal unit, maintained at an ambient temperature, relative humidity of 60% and equal day and night cycles. The animals (*n* = 4 per group) were anaesthetized in an isoflurane chamber with 3–4% isoflurane connected to an oxygen pump for 3–4 min. Intranasal dosing was performed using methods previously described [38]. The mice were left in a separate cage to recover fully after the anaesthesia. Once nanocomplexes were administered, imaging was performed 24 h post-administration, 15 min and 1 h after intranasal dosing with 16 µL of the LuGal substrate (15 mg mL^−1^). 

The in vivo imaging system (IVIS; IVIS^®^-Spectrum systems, Xeno-gen-Caliper Life Sciences, Hopkinton, MA, USA) machine with a cabinet and a CCD camera (2048 × 2048 pixels) was used to image the animals. All anaesthetic procedures were performed with 3–4% isoflurane in the chamber and a maintaining dose of 2% isoflurane during the imaging process in the IVIS cabinet. The system was connected to a computer with a Living Image^®^ 3.0 software (Xeno-gen-Caliper Life Sciences, Hopkinton, MA, USA). An image sequence of four exposure times was generated (30 s, 60 s, 120 s and 240 s). Comparisons between images were performed for the same exposure time of 240 s. The average radiance or the sum of the radiance (photons/second) of each pixel in the region of interest (cm^2^) was identified with a circular selection divided by the number of pixels (p/s/cm^2^/sr), where p = photons, s = seconds, and sr = steradian is compared between treatments.

### 2.13. Brain Dissection and Homogenisation

Mice (*n* = 4 per group) were killed by a CO_2_ overdose followed by decapitation. For downstream analyses, ~1 mg of tissue was manually homogenised in 20 µL of T-PER reagent, according to the manufacturer’s recommendation. T-PER reagent was supplemented with EDTA-free Halt Protease Inhibitor Cocktail (1 mL in 100 mL of T-PER reagent). After preparing the tissue homogenates, the tubes were centrifuged at 10,000× *g* for 5 min and the supernatants were collected. An aliquot of the supernatant (100 µL) was pipetted onto a white 96-well plate and the LuGal substrate (100 µL) was added to each well. Luminescence was measured on a SpectraMax series M plate reader (Molecular Devices, San Jose, CA, USA). The signal from a control animal was subtracted from all the samples. To normalise the data, the relative luminescent units (RLU) per well were divided by the weight of the samples and converted to RLU mg ^−1^ of tissue.

### 2.14. Statistical Analysis

IC 50 values of Lipofectamine and GC37 were compared using an unpaired *t*-test. Statistical analysis of datasets involving multiple comparisons was performed using an ordinary one-way ANOVA with a Bonferroni correction in GraphPad Prism 7. *p*-values less than 0.05 are given one asterisk (* *p*), *p*-values less than 0.01 are given two asterisks (** *p*), *p*-values less than 0.001 are indicated with three asterisks (*** *p*), while *p*-values of less than 0.0001 are marked with four asterisks (**** *p*).

## 3. Results

### 3.1. Acid Degradation of Glycol Chitosan

A relationship between degradation time and molecular weight (Mw) of glycol chitosan has been established previously [36]. Based on this, the time point for degradation was chosen to be 2 h to obtain a glycol chitosan (GC) polymer with a Mw range between 20,000–40,000 kDa. The Mw of the degraded GC batch was measured to be 37,440 Da (GC37) (Mw = 37,440, Mn = 37 200, Mw/Mn = 1.006), while the non-degraded GC had a Mw of approximately 100 kDa (GC100) as reported by the manufacturer, (Figure 1a), after 2 h and 0 h degradation, respectively. Dialysis was used to remove the excess hydrochloric acid from the degradation mixture. NMR analysis provided information on the structure of GC (Appendix A). The structure of GC is presented in Figure 1a.

### 3.2. Toxicity of the Gene Carrier

As a naturally derived polymer, chitosan is often considered biocompatible by default. Indeed, it is biodegradable in vivo by endogenous enzymes in the body, such as lysozyme, which degrades chitosan to *N*-acetylglucosamine [39], the latter being a building block of biomacromolecules, such as glycoproteins, proteoglycans, glycosaminoglycans (GAGs) and other components of the connective tissues [40]. However, biocompatibility must be considered in relation to the structural parameters of the polymer, dosage form, the route of administration and the intended use.

The present study was focused on investigating the effect of GC37 and LX as gene carriers rather than on the toxicity of these formulations in the U87 cell line. Although it is proven that cytotoxicity is cell type-dependent [41], experiments in one cell line can provide early evidence of the cytotoxicity profile of the carriers. GC37 as a polymer carrier was almost 10 times less toxic than Lipofectamine; the latter used as a positive control in all transfection experiments (IC50 GC37 = 3.41 mg mL^−1^ vs. IC50 Lipofectamine = 0.36 mg mL^−1^, * *p* < 0.05), (Figure 1b).

### 3.3. Polyplex Formation

Polyplex formation is confirmed by DLS where naked β-Gal shows the presence of multiple peaks in the size intensity plot (Figure 2a), while glycol chitosan addition at a 60:1 g g^−1^ mass ratio to DNA results in the formation of a monodisperse population of polyplexes as shown by the intensity size distribution plots (Figure 2b). The scaled-up nanocomplexes for in vivo applications showed an increase in particle size with an increase in the amount of DNA used, an observation which has also been reported by others [42,43,44,45] (Figure 2b and Table 1). A possible explanation of the phenomenon has been provided by Mann et al., who investigated the DNA condensation potential of poly-L-ornithine by atomic force microscopy. The authors concluded that with the increase in DNA concentration, multimolecular condensation is observed as opposed to monomolecular DNA condensation, which is believed to be operational when lower amounts of DNA are used [45]. Uncoated polyplexes presented with a positive charge of +7 mV to +10 mV (Table 1).

### 3.4. HYD Coated Polyplexes

GCPH 0.3 and GCPH 0.5 showed no free hyaluronidase for both intensity and volume size distribution plots (Figure 1c) and a significant drop in charge from +10 mV for non-coated polyplexes (GCP) to about −12 mV for the coated polyplexes (GCPH 0.5, **** *p* < 0.0001) (Figure 1d).

The isoelectric point (pI) of crude HYD isolated from bovine testes is reported to be 5.4 [33]. Theoretically, pH values above the pI of an enzyme will result in a net negative charge for the protein, which is desired for the interaction between the positively charged polyplex and the negatively charged enzyme. Dissolving HYD in 20 mM NaCl at a pH of 12 resulted in a pronounced negative charge for the enzyme, because of the extreme pH when compared to 20 mM PBS at a pH of 6.8 (−26 mV and −4.5 mV, respectively, Appendix A). The presence of particles of about 11 nm with 82% abundance is visible from the intensity size distribution of HYD in 20 mM NaCl. Volume and number size distribution plots showed particles of 6 nm with 100% abundance (Appendix A).

Intensity and volume size distribution plots of 0.3 and 0.5 mg mL^−1^ HYD coated GC37 polyplexes (GCPH 0.3 and GCPH 0.5) showed no free enzyme, unimodal size distribution (Figure 1c) and a decrease in zeta potential from approximately +10 mV (GCP) to about −12 mV (GCPH 0.5), **** *p* < 0.0001, (Figure 1d). A size increase is visible for GCPH ternary complexes with an increase in enzyme concentration. Similarly, to the size data, PDI also increased for both GCPH 0.3 and GCPH 0.5 when compared to GCP (0.272 vs. 0.189, ** *p* < 0.01 and 0.282 vs. 0.189, ** *p* < 0.01, (Figure 1d)). This observation is made by others; Dai et al. showed a zeta potential drop and an increase in size for their multi-component nanoparticles comprised of a polymeric shell, an anti-cancer drug, gelatine-RGD and HYD all assembled in one electrostatic complex [46].

The DLS measurements for GCPH 0.3 and GCPH 0.5 ternary complexes showed that using 0.3 and 0.5 mg mL^−1^ of HYD for both polyplexes resulted in enzyme-coated particles of increased size and a negative charge. A step further in the characterisation of the ternary complexes was to use a quantitative method to estimate the amount of free enzyme, i.e., based on the calibration curve using the HPLC method described in the experimental part (r^2^ = 0.9907). The retention time (RT) of HYD was 1.921 min (Figure 1e). Since the size of GCPH 0.5 ternary complexes was ≥200 nm (Figure 1d), a 0.22 μm filter was used to separate coated particles and free enzyme (size of free enzyme ~6 nm) for analytical purposes. A ternary complex with an increased amount of hyaluronidase (GCPH 1) was also used. The filtrate of free hyaluronidase showed an 87% ± 3% recovery after HPLC analysis with the remaining 13% of enzyme putatively left in the dead volume of the filter. The filtrates of GCPH 0.5 and GCPH 1 were analysed and GCPH 1, but not GCPH 0.5, showed a peak with a RT 1.921 in the chromatogram (Figure 1e). By using the area under the curve, it was quantified that 34% ± 5% or 0.34 mg mL^−1^ of free hyaluronidase was present in GCPH 1 filtrate. Due to the filter dead volume, it is uncertain whether all the remaining enzyme is complexed. However, the absence of an HYD peak in the chromatogram of GCPH 0.5 ternary complexes along with the size and zeta potential data provide strong evidence that at a concentration of 0.5 mg mL^−1^ HYD, GCP polyplexes (0.36 mg mL^−1^ GC) are surface coated with the enzyme. Uncoated polyplexes presented with a positive charge of +7 mV to + 10 mV, while GCPH 0.3 and GCPH 0.3 showed a negative charge of about −12 mV, Figure 1d). HYD coated (GCPH 7) polyplexes are stable for 48 h in aqueous media, proven by both intensity and volume size distribution plots (Figure 2c,d).

### 3.5. In Vitro Hyaluronic Acid Digestion Potential of Polyplexes Coated with Hyaluronidase

We have previously developed enzyme-bound particles, using polymeric vesicles prepared from *N*-palmitoyl-*N*-monomethyl-*N*,*N*-dimethyl-*N*,*N*,*N*-trimethyl-6-*O*-glycolchitosan and *N*-biotinylated dipalmitoylphosphatidlyethanolamine, which were subsequently bound to beta-galactosidase streptavidin [47] and found that enzyme activity was preserved in this system. However, immobilisation strategies of enzymes on nanoparticles have been proven to affect the function of the enzyme because of structural changes during the immobilisation process [48,49]. Although an assembly based on electrostatic interactions does not involve chemical modifications to the structure of the enzyme, a pH alteration could hinder important amino acid residues, which can negatively influence the enzyme function [50]. Therefore, to check the potential of the ternary complexes to digest HA in vitro when compared to the free enzyme, an HA digestion assay was used. All ternary complexes retained approximately 50% of their enzymatic activity when compared to the free enzyme regardless of the concentration used (Figure 3a).

### 3.6. Transfection Efficiency of Polyplexes in U87 Glioma Cell Line

GCP 1 polyplexes containing 6 µg mL^−1^ DNA were transfection competent but were found to be significantly less effective at delivering the reporter plasmid to U87 cells when compared to lipoplexes, LX 1 (1.8 mU/well vs. 2.8 mU/well, **** *p* < 0.0001 (Figure 3b)). GCPH 0.3 and GCPH 0.5 completely lost their in vitro transfection efficiency (delivery of β-Gal DNA to U87 cells). Even a further reduction in the concentration of HYD—GCPH 0.2, showed no active β-Gal enzyme expression in U87 cells, when compared to hyaluronidase-free polyplexes (GCP 1, Figure 3b).

### 3.7. In Vivo Studies

IVIS imaging of animals 24 h post nanocomplex administration at a dose of 0.067 mg kg^−1^ and 15 min after substrate administration showed an intense luminescent signal in the nasal cavity. Interestingly, the animal treated with naked β-Gal DNA showed the highest signal (5.6 × 10^8^ p/scm^3^/sr) with distribution close to the administration site (Appendix A). However, GCP 2 and especially GCPH 7 showed a different distribution pattern of the signal when compared to naked DNA, where the luminescent signal of the animal treated with GCPH 7 appeared to spread to more caudal parts of the brain. (Appendix A). Repeated imaging of the same animals an hour post substrate administration showed that the administration of GCP results in a stronger signal when compared to the animal, which received the LX treatment (1.27 × 10^5^ p/s/cm^3^/sr and 2.4 × 10^5^ p/s/cm^3^/sr, respectively, Figure 4a). The signal from the animal administered with naked DNA was almost at the background level (6.7 × 10^4^ p/s/cm^3^/sr) and clearly localised at the tip of the nasal cavity, close to the administration site (Figure 4a). It was then hypothesized that an increase in the dose would allow for the quantification of gene expression. The experiment was repeated with a β-Gal DNA dose increase to 0.2 mg kg ^−1^ (250 µg mL^−1^). Downstream analysis of the homogenised olfactory bulbs, cortex and cerebellum revealed that 24 h post-administration there is no active β-Galactosidase (β-Gal) enzyme in the olfactory bulbs of the animals with any of the treatments (Figure 4b). By contrast, the animals treated with GCP 3 showed significantly higher levels of active β-Gal enzyme in the cortex when compared to animals treated with both LX 3 and naked β-Gal DNA (*** *p* ≤ 0.001), (Figure 4c).

Interestingly, naked β-Gal DNA resulted in significantly higher levels of β-Gal expression in the cerebellum when compared to both LX 3 and GCP 3 (*** *p* < 0.001).

## 4. Discussion

Neurological disorders are the second leading cause of death and disabilities worldwide [51]. Delivery of drugs to the CNS is a cumbersome task due to the presence of the protective BBB. The BBB blocks the transport of nearly 100% of large molecules to the brain and about 98% of the small molecules [52]. Polymer-mediated gene delivery has still a long way to go before the clinical application becomes widespread. To date, there are no approved polymer-based gene therapeutics on the market and despite seven active clinical trials, none are being developed to target CNS disorders [53]. Although viral vectors are much ahead of the competition, taking over more than 70% of all the active gene therapy clinical trials with thirteen approved therapies, the vast majority of the reports are describing invasive and local administration routes. Exploring intranasal delivery as a non-invasive administration route targeting the CNS and bypassing the BBB has sparked research interest in the area [54]. As stated above, the BBB is a major obstacle to the delivery of therapeutics following systemic application (e.g., intravenous delivery). A growing amount of evidence points unequivocally to an extracellular transport of molecules from the nasal cavity to the olfactory bulb and then throughout the cerebrum via the perivascular pathway [28,30,55]. Therefore, a hyaluronidase coating of the nanoparticle was hypothesized to facilitate the transport of the nanoparticles from the nasal cavity enroute to the brain.

We are motivated to do this work as targeted delivery to the forebrain can offer a significant therapeutic advantage for neurological disorders affecting the cortex including Alzheimer’s disease and frontal glioblastoma. It has also been reported that glioblastoma location correlates with specific genetic mutations in patients with frontal glioblastoma associated with isocitrate dehydrogenase 1 (IDH-1) mutations [56], which cause changes in DNA methylation. Additionally, glioblastoma is characterized by the presence of a stiffened and rigid ECM. The stiffened network of overexpressed hyaluronan and other ECM proteins in solid tumours blocks the transport of macromolecules contributing to yet another barrier to the efficient delivery of drugs [57], antibodies [58], immunotoxins [59] and oncolytic adenoviruses [60]. Therefore, the ECM-digestion potential of GCPHs may possibly confer therapeutic benefit to the polyplexes when used via the nose to brain route for gene therapy in glioblastoma.

A 2 h time point for the acid degradation of glycol chitosan was chosen to obtain a polymer with a molecular weight between 20 kDa and 40 kDa, based on previously established molecular weight dependence of GC on degradation time [36]. Figure 1a shows the structure of glycol chitosan as well as the starting molecular weight of the non-degraded material (GC100). The molecular weight of the glycol chitosan obtained after 2 h of acid degradation (GC37) was 37 kDa, as measured by GPC. Peak assignments of corresponding protons in the structure of glycol chitosan are shown in Appendix A.

Elevated levels of cytotoxicity are often associated with an increased charge density along with a decreased degree of deacetylation for chitosan carriers [61]. The present study showed that glycol chitosan as a gene delivery carrier is 10 times less toxic than Lipofectamine (IC50 in the U87MG cell line = 3.41 mg mL^−1^ vs. 0.36 mg mL^−1^, respectively, * *p* < 0.05), Figure 1b.

Polyplexes with three different concentrations of β-Gal DNA were prepared (GCP 1, GCP 2 and GCP 3 containing 6 µg mL^−1^, 84 µg mL^−1^ and 250 µg mL^−1^ DNA) with a resulting size of the nanocomplexes of 135 ± 21, 482 ± 36 and 863 ± 28, respectively (Figure 2b and Table 1). GCP 2 and GCP 3 showed a monodisperse population of particles as determined by DLS with PDI values of <0.2 while GCP 1 showed a PDI value of 0.229. (Table 1). By contrast, unformulated β-Gal DNA presented with a high PDI (>0.5) and three distinct peaks as visible from the intensity size plot (Figure 2a and Table 1).

DLS measurements revealed that GCPH 7 ternary complexes presented with unimodal size distribution and a z-average mean size of 532 ± 71, but also with high PDI ≥ 0.5, which remained unchanged after 48 h. This provides evidence of the short-term stability of the system (Figure 2c,d). Furthermore, no peak indicating free hyaluronidase is present for both intensity and volume size distribution plots (Figure 2c,d). A positive charge was measured for all GCP nanocomplexes ranging from +7 mV to +10 mV and a negative charge of −10 mV for the ternary complexes with hyaluronidase (Table 1). Nanoparticle surface charge is important for stability and prevents the formation of large aggregates. Although glycol chitosan polyplexes and hyaluronidase coated particles possess a slight positive and negative charge respectively, they were stable as confirmed by the unimodal intensity size distribution plots.

Although HYD coated polyplexes GCPH 0.3 and GCPH 0.5 lost their transfection potential in vitro, they retained half of their potential to digest hyaluronic acid when compared to free hyaluronidase Figure 3a,b. LX 1 lipoplexes appeared to be significantly better transfection agents when compared to GCP 1 polyplexes in U87 cells (**** *p* < 0.0001), Figure 3b. The coated polyplexes differ in their size and charge from the non-coated polyplexes, however, even a further reduction of the amount of HYD (GCPH 0.2) did not restore their in vitro transfection potential. While there is more evidence for a positive relationship between increased cationic charge and transfection efficiency, the effect of polyplex size on transfection potential is not clear. Perhaps a 3D spheroid model would be more appropriate to compare the diffusion of the coated nanoparticles in a mimic of the tumour microenvironment. Improved diffusion in tumour spheroids for the multi-component system, when compared to the control nanoparticles, is reported by Dai et al. Additionally the multi-component electrostatic complex with hyaluronidase retained half of its hyaluronic acid digestion potential when compared to the free enzyme [46]. Using electrostatic interactions for complex assembly affects the enzyme function by potentially hindering important amino acids in the active site of the enzyme but does not result in a complete loss of enzyme activity.

Chitosan’s mucoadhesive properties make it a suitable candidate over other polymeric gene delivery vectors for intranasal delivery [62]. Intranasal administration of siRNA (via chitosan-based siNS1 nanoparticles) which acts by silencing the viral NS1 gene before or after infection with a respiratory syncytial virus (RSV) showed significantly decreased virus titres in the lungs and decreased inflammation compared to controls [63]. In addition to the significant attenuation of the infection, siNS1 delivered by the low molecular weight oligomeric chitosan induces 4-day protection from RSV infection emphasizing both the prophylactic and therapeutic potential of the nanoparticles. We have previously shown successful delivery of siRNA in the olfactory bulb neurons following nasal administration using ethyl-amino glycol chitosan polyplexes [64]. The nasal route, rather than the conventional delivery to the upper respiratory tract or the lungs, has also been used for the administration of chitosan-mediated gene therapy for brain delivery as described by Ramos et al. [65]. The authors report on the intranasal delivery of Mn2+ incorporating chitosan nanoparticles (MNPs) carrying a dsDNA coding for a red fluorescent protein (RFP). MNPs of 122 nm in size were shown to deliver the highest amount of RFP, which was detected in the cortex, striatum and hippocampus, with the highest protein expression in the striatum 48 h post intranasal administration.

The perivascular hypothesis for the transport of molecules through the olfactory neuroepithelium, unprotected by the BBB is gaining popularity as the leading mechanism of transport from the nasal cavity to the brain. The proof-of-concept studies of intranasally administered peptides, proteins, and polysaccharides discussed earlier show rapid delivery to brain regions, which do not correlate with the intracellular transport option. However, for gene expression or silencing, more time is needed for an effect to take place. IVIS imaging 24 h post nasal administration and 15 min after substrate delivery show the strongest luminescent signal residing predominantly in the nasal cavity for naked β-Gal DNA (Appendix A). GCP 2 and especially GCPH 7 show the presence of transfected cells in more caudal parts of the brain with the signal spreading further from the administration site (Appendix A). These observations are also confirmed by the later imaging time point of 1 h. GCPH 7 appeared as the most effective gene carrier in vivo with the highest luminescent signal measured at 4.7 × 10^5^ p/s/cm^3^/sr, Figure 4a. GCP polyplexes showed higher signal levels when compared to LXs (2.4 × 10^5^ vs. 1.27 × 10^5^ p/s/com^3^/sr), while naked DNA was at baseline levels (6.7 × 10^4^ p/s/cm^3^/sr) and appeared only at the very tip of the nasal cavity, suggesting minimal transport to the brain. By contrast, GCP 2, LX 2 and GCPH 7 showed signal localization further away from the administration site, Figure 4a. Similar findings are reported by others, where complexed DNA results in longer-lasting gene expression when compared to naked DNA [66]. Tissue homogenization and downstream analysis of the olfactory bulbs, cortex and cerebellum of four animals per group for the four treatments (GCP 2, LX 2, naked β-Gal DNA and GCPH 7, all at 84 µg mL^−1^ or 0.067 mg kg^−1^) resulted in inconclusive data, potentially because of the low dose administered (data not shown). Dose precision in intranasal administration is technically difficult and limiting, as a result of mucociliary clearance and the different anatomies of individual nasal cavities [67]. An increase in the dose of β-Gal DNA at 250 µg mL^−1^ (0.2 mg kg^−1^) and downstream analysis of tissue homogenates of three brain regions (olfactory bulbs, cortex and cerebellum) at 24 h post intranasal dosing showed no active β-Gal enzyme in the olfactory bulbs of the animals (Figure 4b). Significantly higher levels of active β-Gal enzyme (Figure 4c) were detected in the cortex of animals treated with GCP 3 polyplexes when compared both to the naked plasmid (*** *p* ≤ 0.001) and to LX 3 (*** *p* ≤ 0.001). Interestingly, animals treated with naked β-Gal DNA showed the highest protein expression in the cerebellum (Figure 4d) when compared to both lipoplexes and polyplexes (GCP 3 and LX 3, *** *p* < 0.001). A possible explanation for the observed differences is transport through the trigeminal pathway. Olfactory neurons connect the olfactory region of the nasal cavity to the brain—olfactory bulb and frontal cortex. By contrast, the trigeminal neurons connect the respiratory region to more caudal parts of the brain—pons, medulla and spinal cord [68]. However, it is hard to differentiate if carriers/delivery vehicles/substances/materials are using the olfactory or the trigeminal pathway, since the olfactory region is also innervated by branches of the trigeminal nerve, and so it is generally believed that both are involved. Moreover, trigeminal nerve branches passing through the cribriform plate (a horizontal segment forming the roof of the nasal cavity) were shown to be involved in the delivery of substances from the nose to the forebrain [69].

The delivery of the glycol chitosan polyplexes (GCP) to the cortex using a non-invasive route of administration such as the nose to brain route offers great potential for targeting neurological disorders in the cerebral cortex. A recent study by Ramos et al., showed effective silencing of the gene responsible for the synthesis of mutant huntingtin protein (HTT) where HTT mRNA was reduced by over 50% in the olfactory bulbs, hippocampus, striatum, and cerebral cortex at a dose of 5.8 nmol after 48 h [70]. GCP nanoparticles do not need manganese for their preparation and manganese could contribute to the toxicity of the carrier system as the authors suggest [65].

## 5. Conclusions

In summary, we have shown that glycol chitosan polyplexes upon intranasal administration are able to deliver β-Gal DNA plasmid predominantly to the brain cortex and that glycol chitosan is significantly less toxic when compared to the commercial transfection reagent, Lipofectamine. This important finding may be studied further to create gene therapy interventions for neurodegenerative and brain cancer conditions specifically requiring cortex targeting. Naked DNA is preferentially delivered to the cerebellum following intranasal administration and we hypothesize that this is because it utilizes the trigeminal pathway to transport to the deeper brain. Coating polyplexes with hyaluronidase results in gene expression spreading to wider parts of the brain upon intranasal administration, when compared to uncoated polyplexes. However, further studies are needed to prove the role of hyaluronidase coating to facilitate the extracellular transport of molecules from the nasal cavity to the brain.

## Figures and Tables

**Figure 1 pharmaceutics-14-01136-f001:**
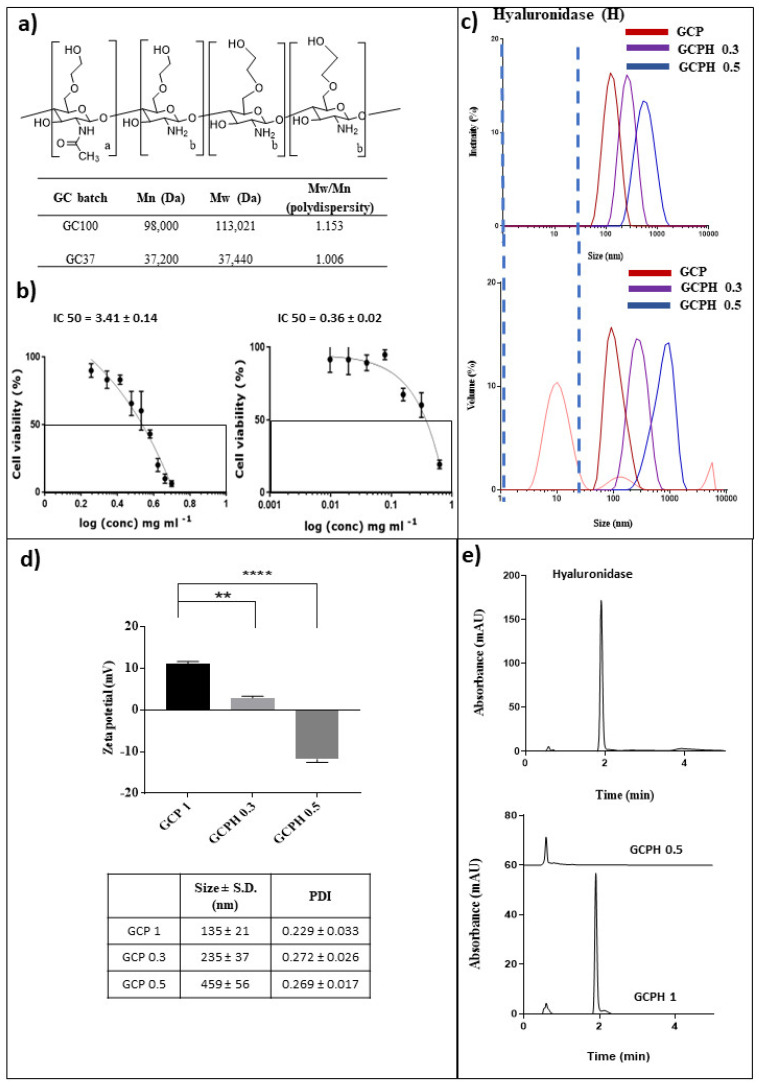
Structure of glycol chitosan (GC), molecular weight and PDI for degraded and non-degraded GC (**a**). IC 50 values for GC37 and Lipofectamine in U87 cells (**b**). Intensity and volume size distribution plots of GCP 1, GCPH 0.3, GCPH 0.5 and hyaluronidase (**c**). Zeta potential values, size and PDI of GCP, GCPH 0.3 and GCPH 0.5 (**d**). The zeta potential of polyplexes without HYD (GCP) is significantly different when compared to polyplexes coated with HYD (GCPH 0.2, ** *p* ≤ 0.01) and (GCPH 0.3, **** *p* ≤ 0.0001) (**d**). HPLC chromatograms of free hyaluronidase (top) and ternary complexes with 1 mg mL^−1^ hyaluronidase-GCPH 1 and with 0.5 mg mL^−1^ hyaluronidase-GCPH 0.5 (bottom) (**e**).

**Figure 2 pharmaceutics-14-01136-f002:**
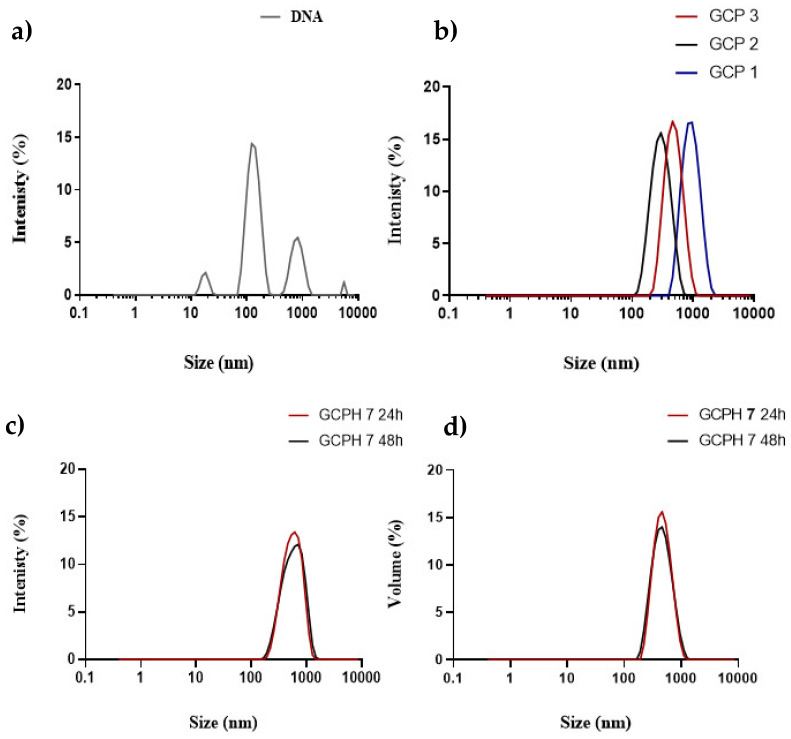
Representative intensity size plots of naked β-Gal DNA (**a**), GCP (GCP 1 contains 6 µg mL^−1^ β-Gal DNA, GCP 2 contains 84 µg mL^−1^ β-Gal DNA, GCP 3 contains 250 µg mL^−1^ β-Gal DNA, all GC, DNA ratios are 60, 1 g g^−1^ (**b**), GCPH 7 (84 µg mL^−1^ β-Gal DNA), with 7.2 mg mL ^−1^ of hyaluronidase size distribution by intensity (**c**) and volume (**d**) at 24 and 48 h.

**Figure 3 pharmaceutics-14-01136-f003:**
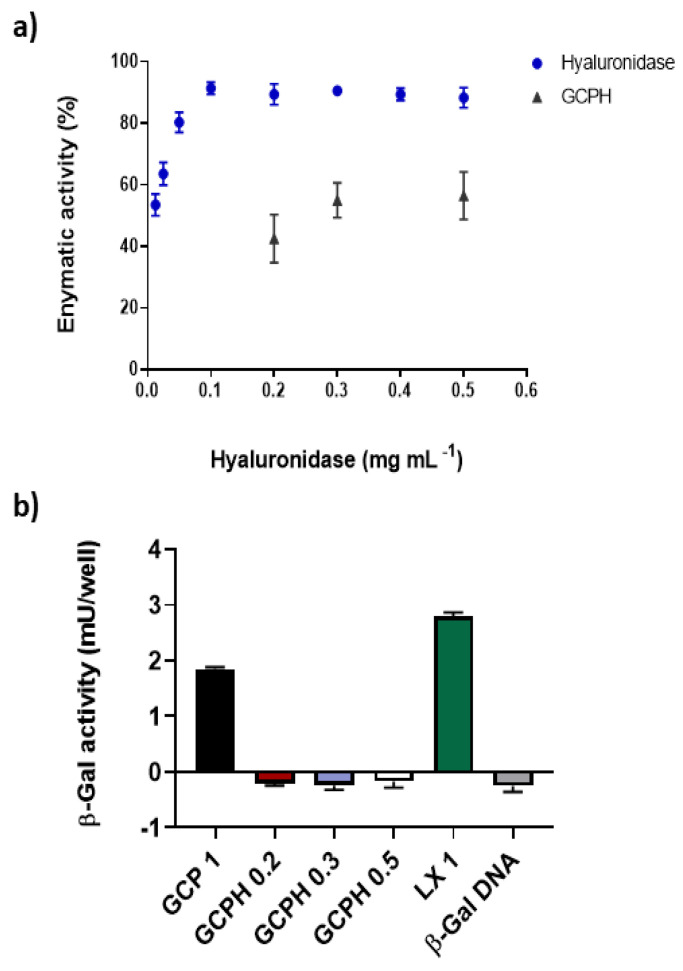
Hyaluronic acid digestion potential of the ternary complexes and free enzyme (**a**). Transfection efficiency of GCP 1 (black bar), GCPH 0.2 (brown bar), GCPH 0.3 (blue bar), GCPH 0.5 (white bar), LX (green bar) and naked β-Gal DNA (grey bar) (**b**). The data is representative of three independent experiments (*n* = 3, mean ± SD).

**Figure 4 pharmaceutics-14-01136-f004:**
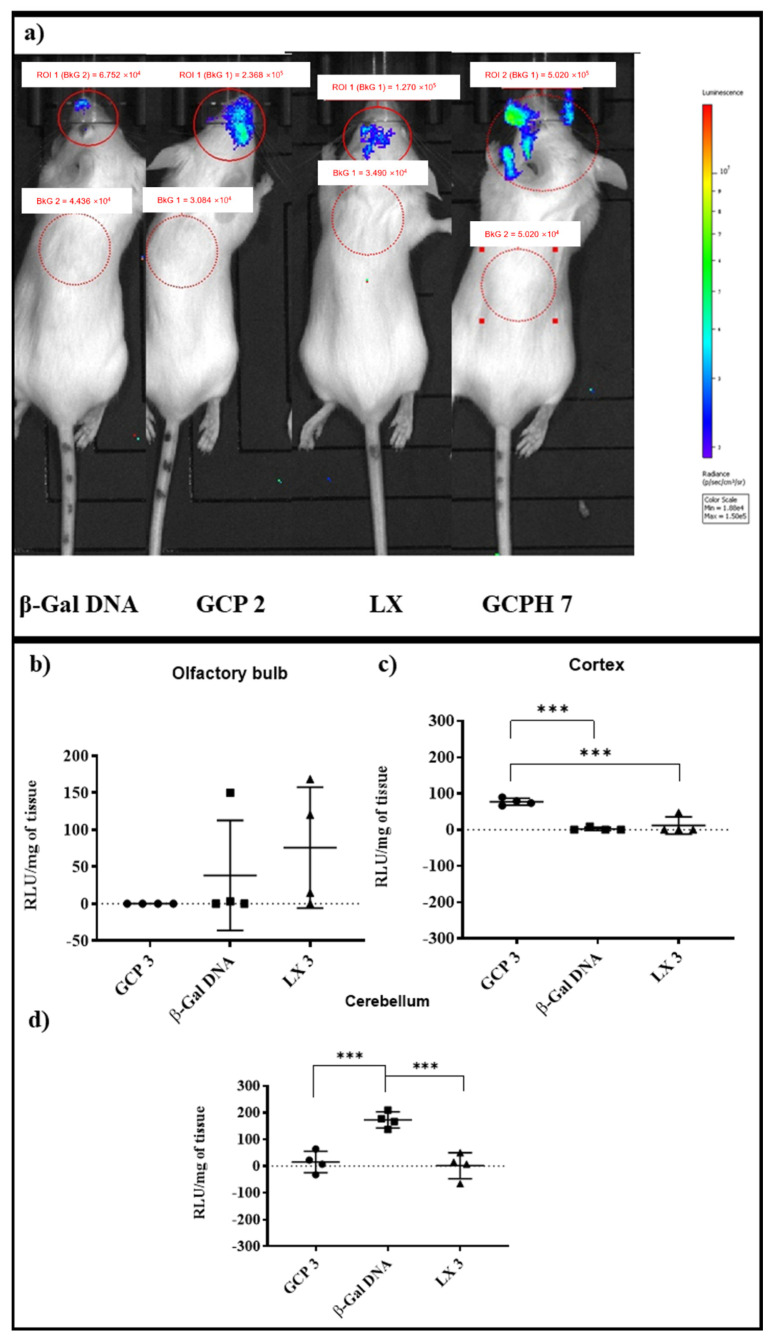
Representative IVIS images from an in vivo intranasal study with nanocomplexes prepared with 84 µg mL ^−1^ DNA or a dose of 0.067 mg kg ^−1^ DNA (**a**). Imaging was taken 24 h post intranasal administration of the treatments and 1 h after intranasal administration of the substrate and with a 240 s exposure time. Downstream analysis of brain homogenates (mean ± s.d., *n* = 4) for active β-Gal expression 24 h post nanocomplex administration at a dose of 250 µg mL^−1^ DNA or a dose of 0.2 mg kg^−1^ DNA in olfactory bulbs (**b**), cortex (GCP 3 polyplexes show significantly higher levels of active β-Gal enzyme when compared to naked β-Gal DNA and LX 3 (*** *p* ≤ 0.001), (**c**) and cerebellum (naked β-Gal DNA shows a significantly higher level of β-Gal enzyme expression when compared to both GCP 3 and LX 3, *** *p* ≤ 0.001). (**d**) The signal from tissue homogenates of olfactory bulbs, cortex and cerebellum of control animals was subtracted from the treatments.

**Table 1 pharmaceutics-14-01136-t001:** Size, PDI and zeta potential of GCP, GCPH and DNA. The data is representative of three independent measurements (=3, mean ± SD).

Polyplex	Size (nm)	PDI	Zeta Potential (mV)
GCP 1	135 ± 21	0.229 ± 0.033	+7 ± 3
GCP 2	482 ± 36	0.113 ± 0.023	+9 ± 4
GCP 3	863 ± 28	0.161 ± 0.021	+10 ± 3
GCPH 7	526 ± 14	0.578 ± 0.053	−10 ± 2
DNA	140 ± 31 (65%)	0.589 ± 0.077	−26 ± 5
	950 ± 61 (23%)		
	20 ± 9 (7%)		

## Data Availability

The data supporting the findings of the study are available from the corresponding author, upon a reasonable request.

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
