# Peer review of "Gene Targeting to the Cerebral Cortex Following Intranasal Administration of Polyplexes"

_pharmaceutics, 2022, doi:10.3390/pharmaceutics14061136_

Round 1
Reviewer 1 Report
The manuscript “Gene targeting to the cerebral cortex following intranasal administration of polyplexes” presents the successive stages of work, from polyplexes obtaining to studying their effectiveness in animals. Below are my comments.
Most of the "Introduction" is devoted to data pointing to the merits of intranasal administration of various drugs, which hardly need to be described so extensively. Dates of publications inspire suspicion of the "value" of this description, because only one article is dated 2020 [25], and this article was written by the authors of this peer-reviewed publication. All other articles were published before 2018, except for one review [8].
Obviously, over the past 8 years since the first publication on the successful delivery of the plasmid to the brain using intranasal administration [20], new data have appeared. Taking into account the rapid development of nanobiotechnologies and their application in biomedicine, the authors should present in the "Introduction" the state of modern research in the field of DNA delivery to the brain using intranasal injection.
In the current version, the text of the "Introduction" is too stretched, I recommend the authors to write more compactly.
In my opinion, the text of the "Materials and Methods" section is too detailed and difficult to read. It can be made more clear by simple editing. It is advisable to provide appropriate references instead of known methods description. It is no need to repeat the conditions for cell cultivation; it is enough to note these once. No need to provide detailed description of Agilent Technologies 1200 series chromatographic system, it is described in the manufacturer's instructions, and so on.
Undoubtedly, the main result of the work is successful delivery of DNA by polyplexes to the mouse brain after intranasal administration. This result is confirmed by IVIS images of animals 24 h post drug administration, as well as the registration of active β-Gal expression at the same time in the homogenates of brain different parts (Figure 5).
However, I cannot accept these data as completely evidential. In similar works, as a rule, they give a series of pictures, especially if there is a different position of animal head. For example, in an animal with the introduction of β-Gal DNA (leftmost) it is clearly visible that the label is localized only in the nose. Authors should present additional panels of IVIS images. Then, if everything is happening as quickly as described in the "Introduction", then why is the data from earlier dates not presented and discussed?
Studies of homogenates should be combined with histological and fluorescent microscopic data, showing exact localization of the label. The brain consists of different tissues, and if we are talking about the targeted delivery of nanoconstructions, we need to use adequate evaluation systems. Homogenates are too crude a system to evaluate directed delivery.
In general, the work showed the possibility of delivering the β-Gal DNA plasmid to the brain using glycol-chitosan polyplexes applied in nasal cavity. I think that this result should be improved, as mentioned above, for a clearer localization of the label in brain. I am sure that the authors of the work have such an opportunity, and then the work will be convincing.
"Discussion" is too long and intricate. I highly doubt that a reader interested in understanding whether the authors have succeeded in delivering DNA by polyplex will read these few pages. At the same time, it would be interesting to discuss the movement of polyplexes in the brain at the cellular and tissue levels, taking into account the peculiarities of the morphological organization of the brain.
I suggest “Major revision” of this manuscript.
Reviewer 2 Report
The manuscript described gene delivery to the cerebral cortex across the olfactory epithelium based on intranasal administration. The authors showed that intanasally administered hyaluronidase coated glycol chitosan – DNA polyplexs (GCPH) were delivered to the cerebral cortex in in vivo assay using mice. Thus, these findings will be useful for drug delivery into the brain. Therefore, the manuscript is not too excellent to be published. In other words, the manuscript is so excellent that it should be published.
Comments
(1) How was mucopermeability of the GCP formulation? Did hyaluronidase (HYD) induce the enhancement of mucopermeability?
(2) How did the GCP formulation cross the apical membrane of the olfactory epithelial cells, receptor-mediated endocytosis or lipid raft-mediated micropinocytosis?
(3) Or did the GCP formulation enter through tight junctions based on paracellular pathway?
(4) Where and when was a reporter plasmid DNA released from the GCP formulation?
(5) How was the effect of mucous flow on the GCP formulation internalization?
(6) Was intranasal naked DNA degraded by DNases?
(7) Was intranasal naked DNA mainly transported, not only across the olfactory epithelium of which the ophthalmic nerve innervates the back, but also the respiratory epithelium of which the maxillary nerve innervates the back (Figure 1. in Molecules 2020, 25, 5188; doi:10.3390/molecules25215188)? Hyaluronidase and mucous flow were thought to be important in mucopermeability.
That is all.
Round 2
Reviewer 1 Report
Dear authors, I am satisfied with your answers and the corrections made to the text.
Let me recommend for the future to introduce microscopy into the set of methods, which will allow even more accurate visualization of the label.